


# Model Comparisons for Predicting Grassland Fire Occurrence Probability in Inner Mongolia Autonomous Region, China

Chang Chang[1,2], Yu Chang[1, *], Zaiping Xiong[1], Xiaoying Ping[1,2], Heng Zhang[3], Meng Guo[4],Yuanman Hu[1,5]

[1]CAS Key Laboratory of Forest Ecology and Management, Institute of Applied Ecology, Chinese Academy of Sciences, Shenyang 110016, China
[2]University of Chinese Academy of Sciences, Beijing 100049, China
[3] College of Forestry, Inner Mongolia Agricultural University, Hohhot 010019, China
[4]School of Geographical Sciences, Northeast Normal University, Changchun 130024, China
[5]E'erguna Wetland Ecosystem National Research Station, Inner Mongolia, 022250, China

*Correspondence to*: Yu Chang (changyu@iae.ac.cn)

**Abstract.** Grassland fires threaten grassland ecosystem, human life and economic development greatly. However, there are limited researches focusing on grassland fire prediction, thereby it is necessary to find a better method to predict probability of grassland fire occurrence. Here we selected 16 environmental variables that may have impacts on fire occurrence, then built regression models of grassland fire probability based on historical fire points and variables in Inner Mongolia by three methods to identify the grassland fire drivers and to predict fire probability. The three methods were global logistic regression, geographically weighted logistic regression and random forest. The results showed that random forest model had the best predictive effect. The influence of 9 variables selected by geographically weighted logistic regression model on grassland fire was unbalanced spatially. The three models all showed that meteorological factors were of great importance to grassland fire occurrence. In Inner Mongolia, grassland fires occurred in different areas had various responses to the influencing drivers, and the areas with different natural and geographical condition had different fire prevention periods. Thus, the grassland fire management strategy based on local conditions should be advocated.

## 1 Introduction

Grassland ecosystems are precious natural resources and have crucial ecological functions, such as maintaining water, regulating local climate condition, providing food and herbs (Steiner et al., 2020; Jiang et al., 2018). Grassland ecosystems are known as an important green ecological safeguard (Pan et al., 2018). Fire is a common disturbance factor in grassland ecosystems (Thomson et al., 2020; Bond and Keeley, 2005). In long term evolutionary process, grassland ecosystems have established a harmonious and balanced relationship with fire. Fire has become an integrated part of grassland ecosystems (Lamont and He, 2017). Fire promotes grassland regeneration, succession, species evolution, and maintains biodiversity in ecosystems (Chandra and Bhardwaj, 2015). However, in the context of global climate change along with human activities, fire frequency is increasing and fire seasons are lengthening (Jolly et al., 2015). Fire has become a threat to biodiversity and





ecosystem stability in the last decades (Verbesselt et al., 2006). Grassland fire is also a sudden and highly destructive disaster that can not only alter the structure, function, pattern and processes of the landscape, but also affect the carbon cycle in grassland ecosystems (Yu and Zhuang, 2019). Besides, it poses threats to herdsman's lives, infrastructures, and valuable

grassland resources (Podur et al., 2003). Prevention of grassland fires has become the main task of local governments to protect herdsman's lives and infrastructures. Accurate prediction of grassland fire occurrences could help to locate high fire risk areas for fire prevention and fighting.

Prediction of grassland fire occurrence needs to properly choose independent variables. With human population increasing and

development of social economy, factors affecting grassland fires are becoming more complicated (Alexandre et al., 2015). Fire occurrence is the consequence of a combination of numerous anthropogenic and natural factors. Among natural factors, meteorological factors are commonly treated as the fire drivers (Oliveira et al., 2012), while vegetation condition or fuels load has important impact on grassland fire burning and spread. Some scholars added variables such as Spring NDVI, Autumn NDVI (Wu et al., 2020a), vegetation condition index (VCI) (Zapata-Rios et al., 2021), percentage of grassland, shrubland,

forest area (Pavlek et al., 2017) that can reflect the vegetation condition or fuels load into the analysis of fire occurrence pattern. In addition, although topography factors show a relatively weaker contribution on fire occurrence pattern, scholars still preferred to select few variables such as topographic roughness index, elevation, slope, aspect index, and surface curvature in fire occurrence prediction (Su et al., 2021). Regarding for the anthropogenic factors, distance to nearest road and settlement is often used as proxy for the intensity of human activities (Liu et al., 2012) for the closer to road and settlement, the more human

activities. Besides choosing variables, appropriate models are also needed to predict grassland fire occurrences.

Over the last decades, a lot of prediction models were developed to explore the relationship between fire occurrence and independent variables (Miranda et al., 2012; Wu et al., 2014). Global logistic regression (GLR) model is the most widely used conventional method on predicting probability of wildfire occurrence (Guo et al., 2016a). It assumes that the interaction

between fire occurrence and its drivers is constant and stable in space. However, due to the environmental heterogeneity in different spatial locations, the relative importance of factors affecting fire occurrence may be different inevitably. This phenomenon is called spatial non-stationarity (Fotheringham, 2002). To solve this problem, geographically weighted logistic regression (GWLR) model was adopted to predict fire occurrence considering the different roles of variables across various spatial locations. GWLR model can deal with the various relationships between dependent variable and independent variable

across geographical locations (Liang et al., 2017). Recently, to improve prediction accuracy, a variety of machine learning methods, including neural networks, support vector machines and random forests (Rodrigues and De La Riva, 2014; De Vasconcelos et al., 2001; Phelps and Woolford, 2021b) also have been used in fire occurrence prediction, among which Random Forest (RF) is deemed to be a flexible method to assess complex interactions among variables for it can automatically select the important variables, regardless of how many variables are input at the beginning, and can overcome the problem of

over-fitting (Guo et al., 2016a; Oliveira et al., 2012). Although GWLR overcomes the shortcomings of GLR, and new machine

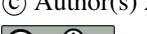


learning methods were proposed, model comparisons are commonly taken to determine the best-fitted model in fire prediction (Mohajane et al., 2021; Phelps and Woolford, 2021a). For examples, Guo (2016) and Oliveira (2012) both compared the GLR model to RF model in fire predictions, and found that RF model performs better than GLR model. Liang had compared the RF model and GWLR model, then got the conclusion that GWLR model had the better performance in forest fire forecasting
Fujian province (Liang et al., 2017). Phelps and Woolford employed the logistic regression, bagged classification trees, random forest, neural network to predict the fire occurrence, however, they found all the models were unsuitable for fire occurrence prediction in their study area (Phelps and Woolford, 2021a).

China possesses nearly $3.92 \times 10^6$ km$^2$ area of grasslands, accounting for 12% of the world's grasslands and 41.7% of the
national land area. Grassland is the main land cover type in Inner Mongolia Autonomous Region (Liu et al., 2018). Meanwhile, the Inner Mongolia grassland is the main component of the temperate grassland of Eurasia, where grassland fire is extremely active (Le Page et al., 2008). Till now, plentiful scholars always paid more attention to forest fire occurrence (Arnan et al., 2013; Matin et al., 2017; Renard et al., 2012; Wotton et al., 2010) and less to grassland fires. Nevertheless, the driving factors and spatial patterns of grassland fires in Inner Mongolia are currently unclear.


In this study, we aim to (1) explore the temporal and spatial distribution patterns of historical grassland fires from 2000 to 2018 in Inner Mongolia to get a whole view of fire points distribution; (2) predict grassland fire occurrence probability based on fire points and environmental variables by three models to identify the drivers of grassland fire occurrence, and to find the best fitting model by comparing the performances of the three models; (3) apply the three models to check whether they can
predict probability of grassland fire occurrence in Inner Mongolia in 2014 accurately and put forward some suggestions on grassland fire management. Our results will facilitate formulating region-specific strategies for grassland resource protection and fire forecast, warning, as well as management. The paper is organized as follows. The second part introduces the general situation of study area, data source, data processing and study methods. In the third part, we present the study results including temporal and spatial distribution patterns of fire occurrences in study area, three regression models and their evaluation as well
as applying the three models. The fourth part discusses the results and their implications for grassland fire management. The fifth part concludes the content of the study.

## 2 Method

### 2.1 Study area

Inner Mongolia autonomous region locates in the north of China (37°24'—53°23'N, 97°12'—126°04'E), and bordered by
Mongolia and Russia. The distance between the east and the west is about 2400 km, and the span between the north and the south is about 1700 km. It has a long and narrow shape extending from the northeast to the southwest with total area of





118.3×10⁴km² (Fig.1a). There are Inner Mongolia plateau, Greater Khingan Range, Hetao Plain in study area. According to the Seventh National Census in China in 2020, the resident population of the district totalled 24.05 million.

Inner Mongolia belongs to temperate continental monsoon climate with the annual average temperature ranging from 0 to 8℃ and annual precipitation from 50 to 450mm (Jia et al., 2020). The climate is characterized by large temperature difference, long sunshine duration, limited and imbalance precipitation. From northeast to southwest, it comprises temperate humid zone, semi-humid zone, semi-arid zone, arid zone and extreme arid zone (Li and Feng, 2019). The grassland vegetation in Inner Mongolia has a distinct zonal distribution pattern from northeast to southwest as well. Three main types of grassland are

temperate meadow grassland, temperate typical grassland, and temperate desert grassland (Zhou et al., 2016) (Fig.1b).

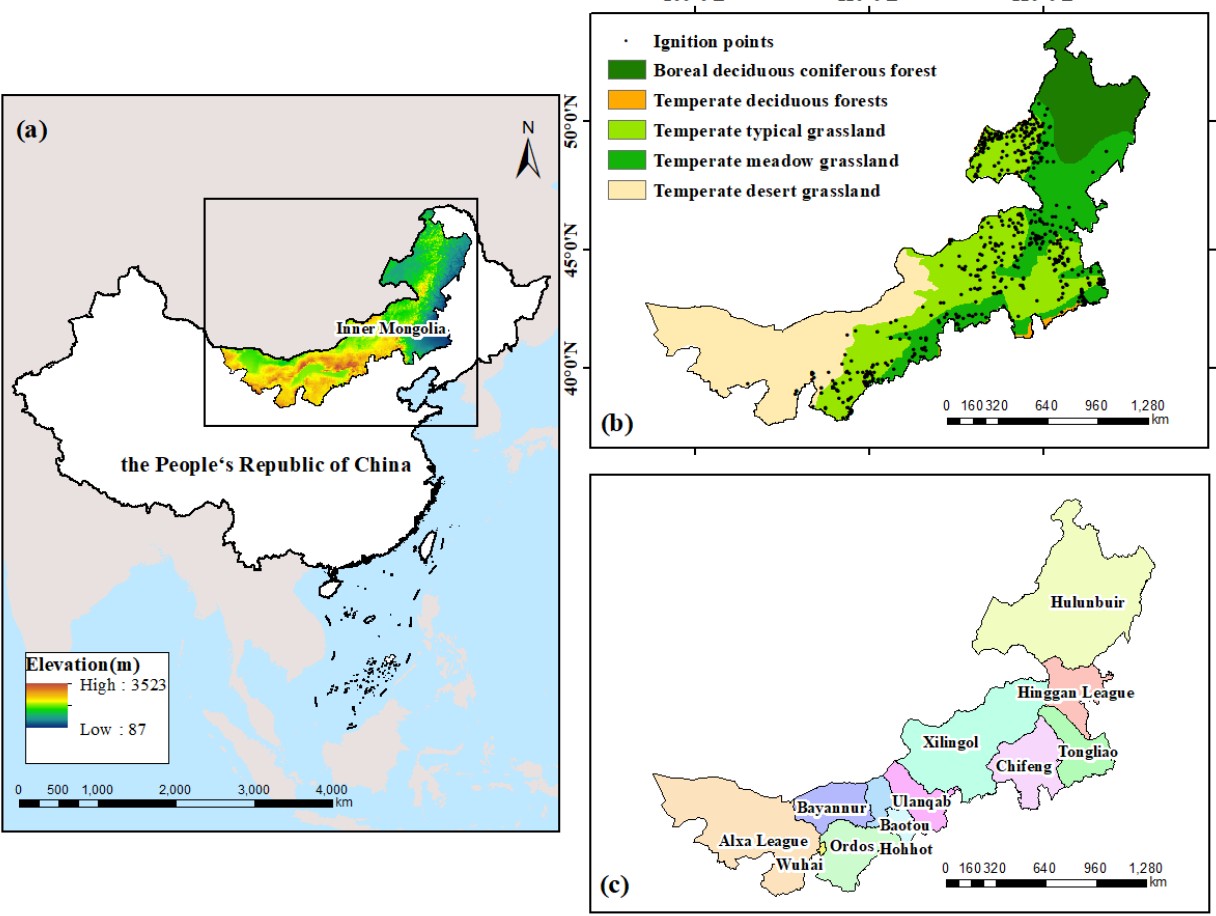

**Figure 1: Geographic location of the study area. (a)The relative position and elevation of Inner Mongolia, (b)fire ignition points and vegetation type map, and (c) administrative map on the city level.(All the base map layers except ignition points were from © Institute of Geographic Sciences and Natural Resources Research, CAS).**



## 2.2 Data source and pre-processing

### 2.2.1 Dependent variable

We used the historical grassland fire occurrence records as the dependent variable. These fire records came from Inner Mongolia fire management department that contained detailed information of each grassland fire in Inner Mongolia, including geographical location, fire starting time, fire size, fire cause, casualties and so on. There were 1049 fires recorded from 2000 to 2018, of which 685 fires had geographical coordinate information, thus the remains without geographical information were removed. The geographical distribution patterns of fire occurrences were shown in Fig.1b. Based on the 685 valid fire occurrence points, the non-fire occurrence points were created that had the 1.5 times number of fire points (Catry et al., 2009) by ArcGIS10.6. Therefore, the total sampling points were 1711. Then an attribute value of the points was coded 0 for the non-fire points, and 1 for fire points. This attribute was the dependent variable, which requires for coding in a binary format in logistic regression model (Wheeler, 2007).

### 2.2.2 Explanatory variables

Explanatory variables used to predict fire occurrence probability were divided into 4 categories, meteorology, anthropogenic activity, vegetation and topography. All the explanatory variables selected were as follows: mean annual temperature and precipitation, daily average wind speed, daily average temperature, daily average specific humidity, daily cumulative precipitation; distance to nearest settlement, distance to nearest road, distance to nearest river, distance to nearest railway, population density; Normalized Difference Vegetation Index (NDVI), global vegetation moisture index (GVMI); elevation, aspect and slope. The data source, abbreviation and unit of all the variables were listed in Table1.

*Meteorological factors*

Meteorology plays an important role on fire occurrence. Mean annual temperature and precipitation were selected as climate factors because they influence fuel moisture content which is a crucial factor for fire occurrences. In addition, they are traditional indicators for the degree of climate change (Scholze et al., 2006). Daily average temperature, wind speed, specific humidity and cumulative precipitation influence the moisture content of fuels more directly (Shmuel et al., 2022). Meteorological factors in this study came from China Meteorological Forcing Data (1979-2018) (Yang, 2018). NetCDF is the original format of the data, which can be disposed by ArcGIS. Mean annual temperature and precipitation, daily mean wind speed, daily mean temperature, daily mean specific humidity and daily cumulative precipitation, were extracted from this dataset. The value of each variable was extracted by the locations and date of the sampling points. In addition, daily meteorological data of non-fire points was extracted by a random date without fire occurrence in the period of 1990 to 2018.

*Anthropogenic factors*

Human activities are closely related to the human-caused fire occurrence (Vilar et al., 2010). The distance to human infrastructure, such as road, railway and settlement is usually used to represent the intensity of human activity (Liu et al., 2012;



Chang et al., 2013). In Inner Mongolia, herdsmen prefer to feed water for livestock by rivers, so the closer to the river, the greater the intensity of human activity. Based on this, we selected the distance to nearest settlement, road, railway and river as the anthropogenic factor. Digital maps (1:250,000) of railway, road, river, and resident points were collected from National

Catalogue Service for Geographic Information. We calculated the Euclidean distances from each fire location to the nearest railway, road, river, and settlement. These distances were acted as proxies of intensity of human activities. Then the values of these distances were extracted by the locations of the fire records. To some extent, population density showed strong correlation with human activity intensity. Population density data of counties in the whole study area was drawn into raster layers and was extracted by the locations of the sampling points.

*Vegetation factors*

FVC can be used to indicate the gross of live and dead fuels above the surface (Purevdorj et al., 1998). Normalized Difference Vegetation Index (NDVI) has the similarity with FVC, so it can be replaced by FVC under some circumstances. Global Vegetation Moisture Index (GVMI) can provide direct information on vegetation water content at canopy level (Ceccato et al., 2002a) thus reflect the moisture content of live fuels. Therefore, two index, NDVI and GVMI were selected to represent fuel

load and fuel moisture conditions respectively. We calculated the average NDVI value of the years from 2000 to 2018, creating a NDVI layer. GVMI is based on combination of near-infrared and short-wave infrared bands of the electromagnetic spectrum to estimate vegetation moisture content (Ceccato et al. 2002). Band 2 and band 6 of product MOD09A1 of MODIS surface reflectance was used. The method to calculate the GVMI can be found in Ceccato's research (Ceccato et al., 2002b).

*Topographic factors*

The topographic factors affect vegetation distribution, composition and flammability (Syphard et al., 2008). The higher the elevation, the lower the temperature. Different aspect will receive imbalance radiation from the sun. Hence, the vegetation type changes with elevation and aspect. A study found that human-caused fires occurred more frequently in gentle slopes (Conedera et al., 2011). Elevation data came from Geospatial Data Cloud. The 90m spatial resolution digital elevation model (DEM) was adopted, then the processes of spatial correction mosaic, clip, and resample on DEM were indispensable. By the

3D Analyst tool of ArcGIS10.6, raster layers of slope and aspect can be obtained. Aspect needed to be exponentiated, which was convenient for analysis. The aspect was converted into aspect index according to specific method provided by Su (Su et al., 2021).

Explanatory variables could be extracted from the corresponding map layers, the value of each explanatory variable was

extracted as an attribute of the sampling points to form the completed sampling dataset collection, then we randomly chose 60% from fire points and non-fire points respectively as training dataset, and the remained 40% as test dataset.

**Table1 Explanatory variables and their sources**

| Factors | Variables | Abbreviation | Data source | Units | Resolution |
|---------|-----------|--------------|-------------|-------|------------|
| | Mean annual temperature | Temp | | K | 0.1° |



| Meteorological factors | Mean annual precipitation | Prec | China Meteorological Forcing Data (1979-2018) (Yang, 2018) | m/s | 0.1 ° |
| | Daily average specific humidity | Humi_dy | | kg/kg | 0.1 ° |
| | Daily cumulative precipitation | Prec_dy | | mm/hr | 0.1 ° |
| | Daily average temperature | Temp_dy | | K | 0.1 ° |
| | Daily average wind speed | Wind_dy | | m/s | 0.1 ° |
| Anthropogenic factors | Distance to nearest settlement | D_resp | National Catalogue Service for Geographic Information (https://www.webmap.cn) | km | 500m |
| | Distance to nearest road | D_road | | km | 500m |
| | Distance to nearest river | D_river | | km | 500m |
| | Distance to nearest railway | D_rail | | km | 500m |
| | Population density | P_density | National Bureau of Statistics of China (http://www.stats.gov.cn) | per/km$^2$ | 500m |
| Vegetation factors | Global vegetation moisture index | GVMI | Level-1 and Atmosphere Archive & Distribution System Distributed Active Archive Center (https://ladsweb.modaps.eosdis.nasa.gov/) | - | 500m |
| | Normalized Difference Vegetation Index | NDVI | Resource and Environment Science and Data Center (Xu, 2018) | - | 1000m |
| Topographic factors | elevation | Elev | Geospatial Data Cloud (http://www.gscloud.cn) | Meter | 500m |
| | aspect index | Aspect | | - | 500m |
| | slope | Slope | | degree | 500m |

## 2.3 Data Analysis methods

### 2.3.1 Multicollinearity diagnosis between explanatory variables.

If there is strong multicollinearity among explanatory variables, the regression fitting may produce biased parameter estimation, leading excessive standard error of regression coefficient, thus the model will be unreliable (Wheeler, 2007; Chang et al., 2013). Variance inflation factor (VIF) is a common method to evaluate multicollinearity among the variables. It is stated that if VIF>10, there is severe collinearity; 5<VIF<10, there is moderate collinearity; 2<VIF<5, there is mild collinearity (Krebs et al., 2012). Therefore, we did a multicollinearity diagnosis for our sampling dataset by VIF, The VIF test was manipulated by car package of R studio 4.0.3. In this study, the VIF value of variables Temp, Prec, Humi_dy, Prec_dy Temp_dy, Wind_dy, D_resp, D_road, D_river, D_rail, P_density, GVMI, NDVI, Elev, Aspect, Slope were 1.024, 5.292, 1.8434, 1.253, 1.613, 1.140, 1.396, 1.430, 1.507, 1.352, 1.189, 1.103, 5.381, 1.800, 1.032, 1.490 respectively. This indicated that there was moderate



collinearity for Prec variable and NDVI variable with their VIF greater than 5. In consideration of this, we deleted Prec variable

in the global logistic regression and geographically weighted logistic regression to ensure the model reliability.

### 2.3.2 Trend analysis of grassland fires

In order to get a concrete and overall distribution of all the historical fire occurrence points, we need a tool to present them clearly. Trend analysis is an analytical tool in ArcGIS geographic statistics module, which can analyse the spatial distribution trend of data (Mou, 2012). In order to find the spatial distribution patterns of grassland fire occurrences, the Inner Mongolia

map was divided into 6064 grids of 20k×20k by the create fishnet tool, then the fire points in each grid were calculated. Trend analysis in ArcGIS was used to get a three-dimension overview of fire points in geographical space. Additionally, to find the characteristics of time change in a year, sample quantity statistics was done at the half-month interval according to the fire occurrence date from 2000 to 2018.

### 2.3.3 Modelling methods

Regression methods have been widely applied in fire occurrence modelling, such as linear regression (Oliveira et al., 2012) or logistic regression (Phelps and Woolford, 2021b; Chang et al., 2013). These models are often built with the goal of using the fewest predictors to explain the greatest variability in the dependent variable (Graham, 2003). Random forest model has showed its strong predictive ability in fire occurrence probability (Guo et al., 2016b; Oliveira et al., 2012). Threrefore, we selected the logistic regression and random forest model to build our fire occurrence models. Geographically weighted logistic

regression shows its ability on dealing with spatial heterogeneity, so we picked it. The following is an introduction to the three models.

*Global logistic regression (GLR)*

Generalized Linear Models are widespread methods in plenty of research fields and logistic regression is one of the most popular approaches in occurrence modelling in fire science (Bar Massada et al., 2013). When predicting a binary variable

through a series of continuous or categorical predictor variables, logistic regression is a useful tool. LGR assumes that the samples are independent and have a constant relationship between the dependent variable and the independent variables in the entire study area (Zhang et al., 2013). The global logistic regression was performed by AER package in R studio4.0.3.

*Geographically weighted logistic regression (GWLR)*

GWLR firstly assumes that the model structure is spatially non-stationary which means the relationship between the binary

dependent variable and the continuous independent variables changes with geographic locations. Then, it tests whether the spatial non-stationary relationship of this hypothesis is significant. LGWR shares similarity with its global counterpart, producing the $\beta$ regression coefficients and significance tests. It is worth noting that rather than a single parameter, a collection of parameters for each point or polygon can be obtained. Therefore, such model often outperforms global regression models (Fotheringham, 2002). A conventional GWR is described by the following expression:


$$y_i = \sum_k \beta_k(a_i, b_i)x_{k,i} + \varepsilon_i \qquad (1)$$

where $y_i$, $x_{k,\,i}$ and $\varepsilon_i$ are, dependent variable, $k$th explanatory variable, and the Gaussian error at location $i$ respectively; $(a_i, b_i)$ is the X/ Y coordinate of the $i$th location; and coefficients $\beta$ $(a_i, b_i)$ are varying depending on the location.

We used the GWR4 software freely accessible to build GWLR model and chose Adaptive bi-square as kernel type, golden section search as the bandwidth selection method. Generally, the fixed kernel specifies an equal distance threshold for each regression point, while the adaptive kernel specifies the number of neighbours to be considered for each regression point.

Therefore, fixed kernels approach should be suitable in a circumstance where the points are regularly distributed in space and the adaptive approach is more appropriate for spatially clustered points (Nunes et al., 2016). According to the spatial distribution pattern of our fire occurrence data, Adaptive approach was selected to build model. Akaike information criterion (AIC) approach was used to assess the goodness-of-fit for each model.

*Random forest (RF)*

Random forest model is a kind of supervised machine learning algorithms based on decision trees. RF model gathers a large number of classification trees to improve the prediction accuracy of the model. It is unnecessary to set the function form in advance, and also can overcome the complex interactions between covariables to get a high classification accuracy (Gao et al., 2020). It can handle a large number of independent variables and pick out important ones. Besides, it is not affected by the

multicollinearity between variables and can evaluate the complex interactions between variables more flexibly (Ma et al., 2020). Random forest has the advantages of having higher accuracy than individual decision trees and low sensitive to parameter adjustment than other machine learning models (Su et al., 2020). However, random forest model also has the limitation that it can't calculate the specific regression coefficient and confidence interval. We built random forest model by randomForest package in R studio4.0.3.

**2.3.4 Model evaluation methods**

To evaluate the goodness of fit of the three models we chose, four statistical measures, akaike information criterion (AIC), area under curve (AUC), mean absolute error (MAE) and $R^2$, were selected. AIC is based on the concept of entropy, which can weigh the complexity of the estimated model and the goodness of model fitting data. It is an index obtained in the process of model fitting. The receiver operating characteristic (ROC) curve was obtained by plotting sensitivity versus specificity for

various probability thresholds. The area under the curve (AUC) is also often used to evaluate model performance (Jiménez-Valverde, 2012). MAE is the average value of absolute error, which can better reflect the actual situation of predicted value error. MAE is defined in the following expression:

$$MAE = \frac{1}{n}\sum_{i=1}^{n} |\hat{y}_i - y_i| \qquad (2)$$

$R^2$ can reflect the fitting degree of the regression line to the observed value, which can be calculated by the following equation:




$$R^2 = 1 - \frac{\sum_{i=1}^{n}(\hat{y}_i - y_i)^2}{\sum_{i=1}^{n}(\hat{y}_i - \bar{y}_i)^2}$$
(3)

where n is the number of samples, $\hat{y}_i$ is fire occurrence probability predicted by the model, $y_i$ is a binary value representing whether or not a fire exists, $\bar{y}_i$ is the arithmetic mean of binary values.

## 3 Results

### 3.1 Temporal and spatial distribution patterns of fire occurrences

From a spatial perspective, the fire points gathered in the east and central parts of Inner Mongolia, and the east part had more fire points than the central, while the west part had limited fire points in longitude direction. The fire points distribution was more balanced in latitudinal direction, except for the northernmost part where no fire occurred due to the lack of grassland (Fig.2).

From a temporal perspective, there were two peaks of fire occurrences during a year. The first one appeared in the first half of April, and the second appeared in the first half of October (Fig.3). The number of fires during the first fire prevention period was significantly higher than that during the second fire prevention period. According to the *Regulations on Fire Prevention of Forests and Grasslands in the Inner Mongolia Autonomous Region*, there are two fire prevention periods, March 15th to June 15th, and September 15th to November 15th. We found that the periods when the number of fires was greater than 20
coincided with the fire prevention periods regulated in the region.

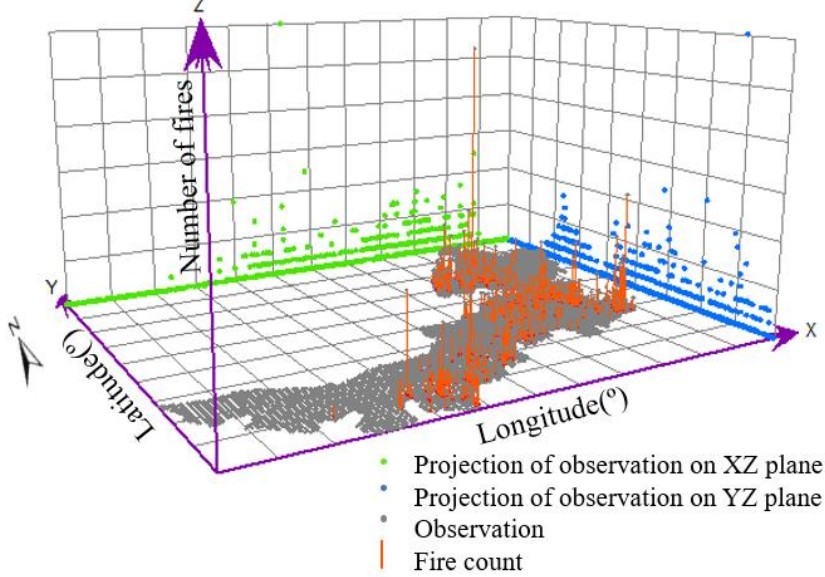





**Figure 2. The spatial distribution of grassland fires in Inner Mongolia from 2000 to 2018. (The height of the orange line represents number of fires; the grey point is the position of the observation point and the projection on the XY plane. The green point is the projection of the observations on the XZ plane; the blue point is the projection of the observations onto the YZ plane)**

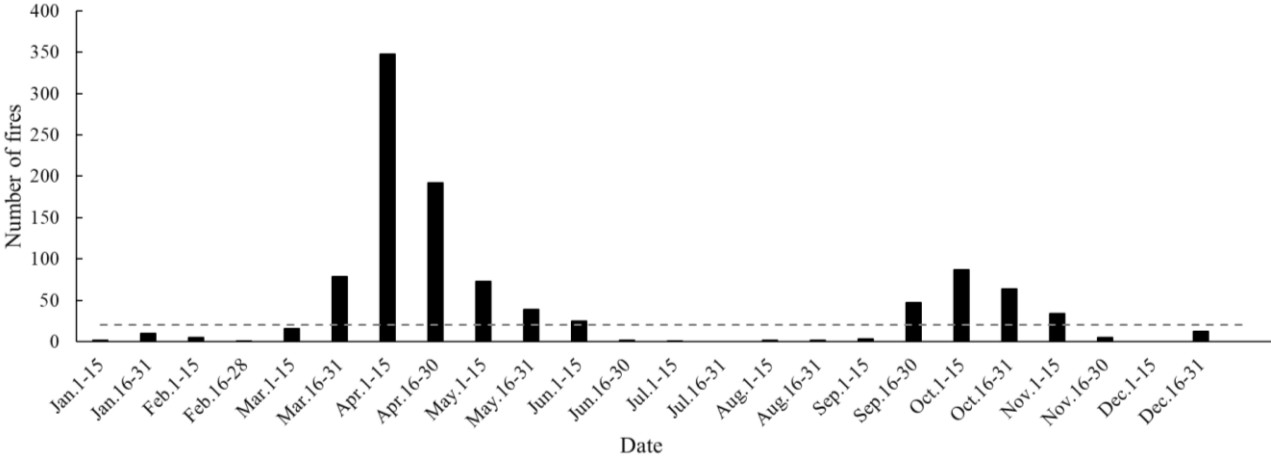


**Figure 3. Changes of the number of grassland fires in Inner Mongolia at the half−month interval. The grey dotted line is the reference line for 20 fires.**

## 3.2 Model fitting and evaluation

Based on GLR model, only 9 variables had significant coefficients (p<0.05) among the 16 explanatory variables, namely Temp,

Humi_dy, Wind_dy, D_river, D_resp, D_rail, GVMI, NDVI and Slope that were regarded as grassland fire drivers. Among the drivers, only NDVI and Wind_dy were positively correlated with fire occurrences, and the rest of drivers were negatively correlated with fire occurrences (Table 2).

**Table 2 Parameter estimation of explanatory variables by the GLR model**

| Variable | Abbreviation | Coefficient | Standard error | P-value |
|---|---|---|---|---|
| Intercept | Intercept | 38.543 | 10.218 | <0.0001 |
| Mean annual temperature | Temp | -0.138 | 0.036 | <0.0001 |
| Distance to nearest river | D_river | -0.009 | <0.0001 | <0.0001 |
| Distance to nearest settlement | D_resp | -0.142 | <0.0001 | <0.0001 |
| Distance to nearest rail | D_rail | -0.004 | <0.0001 | 0.024 |
| Global vegetation moisture index | GVMI | -7.523 | 1.650 | <0.0001 |
| Normalized Difference Vegetation Index | NDVI | 2.599 | 0.560 | <0.0001 |
| Daily average specific humidity | Humi_dy | -857.033 | 81.818 | <0.0001 |
| Daily average wind speed | Wind_dy | 0.217 | 0.050 | <0.0001 |


| Slope | Slope | -0.108 | 0.024 | <0.0001 |
|---|---|---|---|---|

The drivers selected by the GWLR model were consistent with the GLR model. The GWLR model spatialized the coefficient

of each of the selected explanatory variables (Fig.4). These coefficients demonstrated large spatial variations (Table 3).

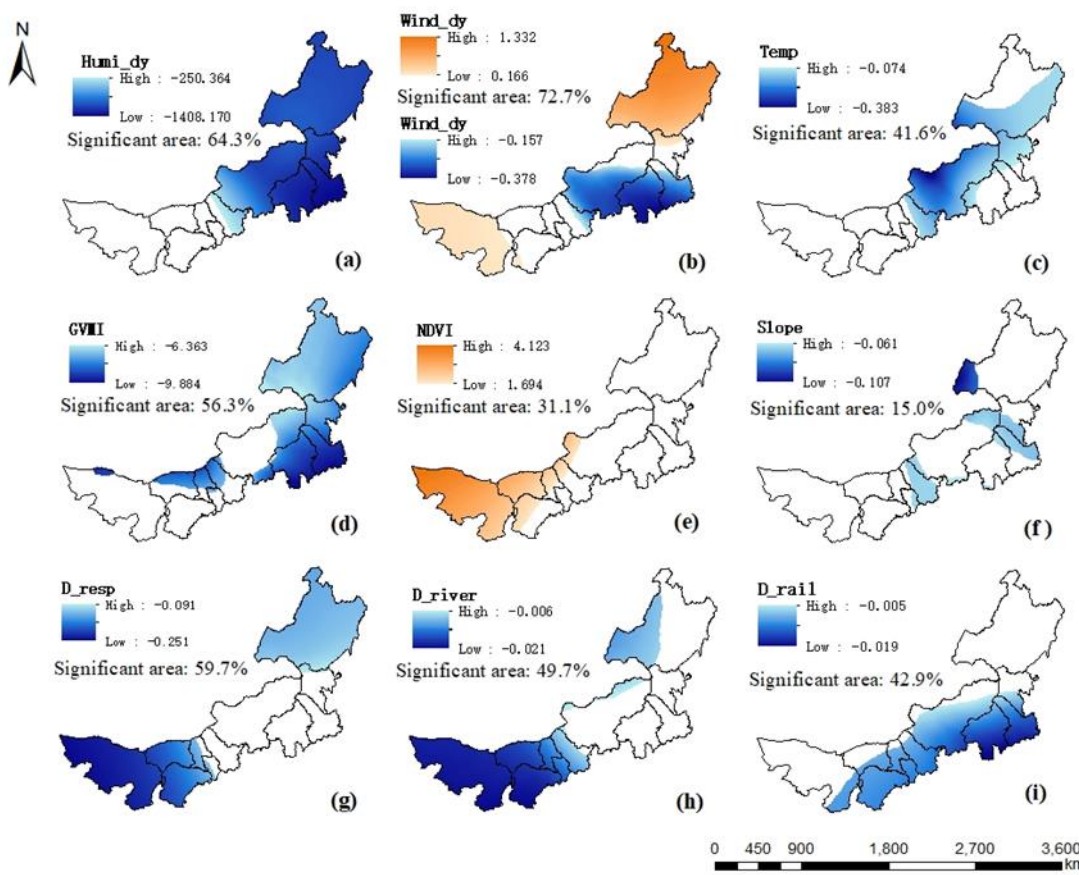

**Figure 4. The spatial distribution of significant areas of the estimated coefficient for each variable selected by GWLR model. These maps were drawn according the t-value for each pixel. The pixels with the corresponding t-value between -1.96 and 1.96 (insignificant**
**area) were not displayed on the graph. Those pixels with the t-value greater than 1.96 or less than -1.96 (significant area) were displayed on the graph. The negative coefficient values were coloured in cool colours, and the positive coefficient values were coloured in warm colours.**

**Table 3  Coefficient statistics of explanatory variables selected by the GWLR model**

| Variable | Mean | Min | 1Q | Median | 3Q | Max |
|---|---|---|---|---|---|---|
| Intercept | 30.55973 | 8.352316 | 14.07005 | 28.78026 | 40.13325 | 109.0789 |
| Humi_dy | -711.947 | -1407.05 | -1115.85 | -1037.32 | -26.1174 | 262.0729 |
| Wind_dy | 0.204153 | -0.37813 | -0.21256 | 0.123083 | 0.534187 | 1.327401 |
| NDVI | 1.294488 | -1.53458 | 0.749144 | 1.351105 | 1.899675 | 4.123084 |





| | | | | | | |
|---|---|---|---|---|---|---|
| GVMI | -7.48481 | -9.86252 | -8.10707 | -7.24877 | -6.92158 | -3.26509 |
| Temp | -0.10685 | -0.38292 | -0.14078 | -0.09896 | -0.0535 | -0.02163 |
| Slope | -0.06249 | -0.10811 | -0.0675 | -0.0622 | -0.05487 | -0.03551 |
| D_rail | -0.00752 | -0.0189 | -0.01085 | -0.00803 | -0.00173 | -0.00097 |
| D_resp | -0.11175 | -0.25096 | -0.14876 | -0.12328 | -0.05028 | 0.016634 |
| D_river | -0.00949 | -0.02133 | -0.0166 | -0.00881 | -0.00415 | 0.002383 |

Among the meteorological factors, the significant area of Humi_dy was distributed in the eastern and central area of the study area, accounting for more than 64.3% of the total area. The positively significant area of Wind_dy located in the northernmost and westernmost area, including Hulunbuir and almost the entire Alxa, the negatively significant areas were concentrated in the middle. Temp was significant in some parts of the east, showing a negative correlation. Among the vegetation factors, GVMI was more significant in the eastern region with small areas distributing in the central and western regions. NDVI

positively significant correlated with fire occurrence in a small area in the west. As for the topographic factors, only Slope was negatively correlated with significant area accounting for only 15% of the study area, scattered in the eastern and central regions. Among anthropic factors, the significant area of D_resp was found in Hulunbuir and the western region, and had a greater impact on fire occurrences in the west. D_river was significantly negatively correlated in the western Inner Mongolia and the west of Hulunbuir. The significant area of D_rail resided in southern Inner Mongolia.


The RF model picked all 16 explanatory variables into the model during the fitting process. In order to understand the importance of each variable in the model, the Mean Decrease Accuracy of IncNodePurity was used to calculate the importance of each variable, then the importance ranking of variables was shown in Fig.5. In general, the importance of each explanatory variable was quite different. The importance of the two daily-scale meteorological variables, Humi_dy and Temp_dy, was

much greater than other variables. In the top 6 of importance ranking, there were five meteorological variables, indicating the meteorological conditions were the main cause of the grassland fire occurrences. The importance of NDVI variable which could reflect the load of fuels, was relatively high in accordance with expectation. Among the three topographical variables, Slope and Aspect were of lower importance, indicating that topographical conditions have less impact on the occurrences of grassland fires in Inner Mongolia.


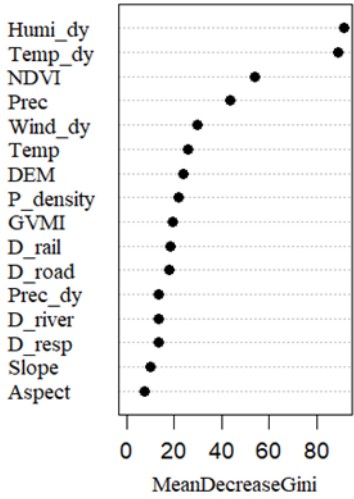


**Figure 5. The importance ranking of the explanatory variables in RF model**

Table 4 listed several statistical measures to evaluate the performance of GLR, GWLR and RF models. The RF model as a machine learning algorithm cannot obtain the AIC value, so only the GLR and GWLR models were compared according to AIC. The AIC value of GWLR model was smaller than the GLR model, which meant GWLR model had a higher goodness of

fit. The AUC value of GLR, GWLR and RF model was 0.841, 0.909 and 0.944; MAE value was 0.148, 0.111 and 0.053; R2 was 0.363, 0.537, and 0.779 respectively. By comparisons we believed that RF model performed better than both GWLR model and GLR model, and could more accurately predict grassland fire occurrences in the region.

Table 4   Comparisons of the fitting goodness of three models

| Model | Akaike information criterion (AIC) | Area under Curve (AUC) | Cut-off value | Mean Absolute Error (MAE) | $R^2$ |
|-------|------------------------------------|------------------------|---------------|---------------------------|-------|
| GLR   | 1018.496 | 0.841 | 0.475 | 0.148 | 0.363 |
| GWLR  | 807.330  | 0.909 | 0.471 | 0.111 | 0.537 |
| RF    | –        | 0.944 | 0.500 | 0.053 | 0.779 |

**3.3 Model application**

According to RF model, daily meteorological conditions were of high importance compared to other explanatory variables. The vast territory of Inner Mongolia and geographical features lead to great spatial variations in meteorological conditions, which would result in greater differences in the probability of fire occurrence in east and west region. Therefore, we randomly chose two points, one in the eastern and the other in the western area. Point 1 was located in the Hulunbuir grassland in northeastern Inner Mongolia, and Point 2 was located in the Ordos Plateau, both of which were areas with a high historical fire




points density. Based on the 19-year fire records obtained, we found that 2014 was the year with the highest number of

grassland fires, with a total of 111 occurrences. For this reason, the year of 2014 was token an example. We used the three

models to estimate their probability of fire occurrence with daily meteorological variables in 2014 and other explanatory

variables.

Fig.6a showed the estimated probability of fire occurrence by the three models at Point 1, and Fig.6c at Point 2 on each day in

2014. We found that the estimated probability of fire occurrence by GLR and GWLR were higher than RF at both locations.

During none fire prevention periods, the probability by GLR and GWLR were also much higher while lower by RF, indicating

that RF model could reasonably capture the fire prevention periods. In addition, we also found that the probability of fire

occurrence estimated by each model at Point 1 was higher than Point 2. This may be due to higher fuel loads at Point 1 than

Point 2.

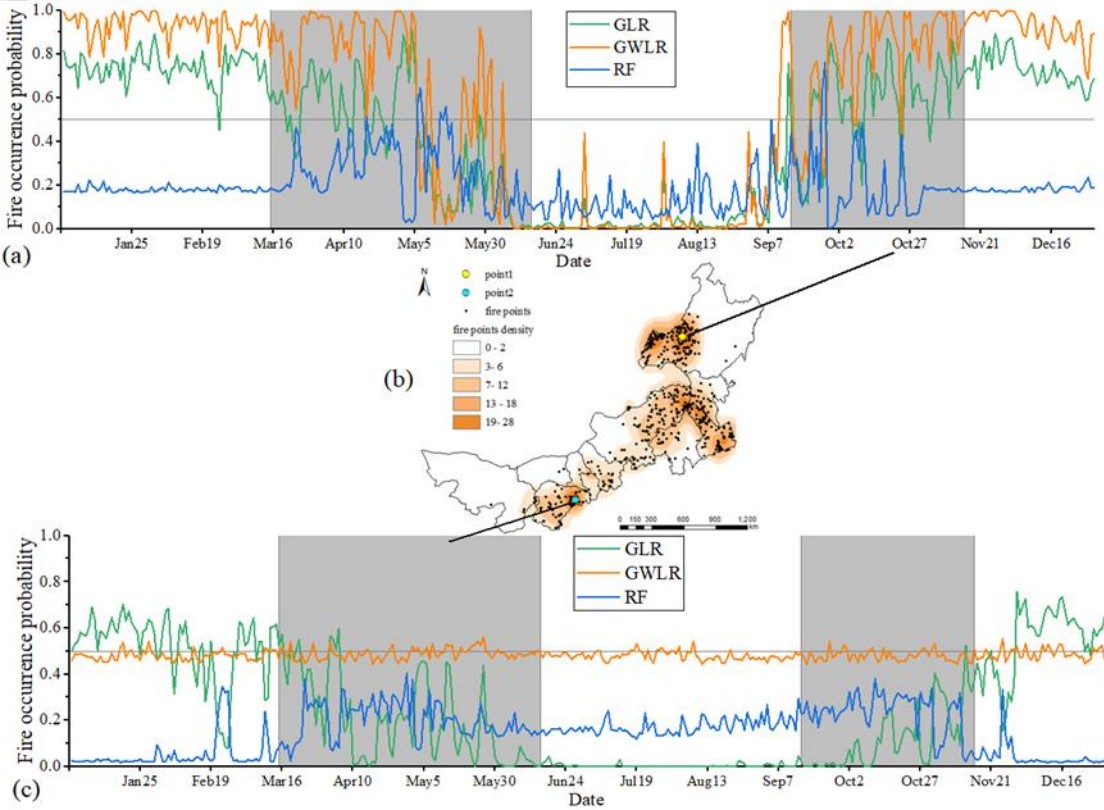

**Figure 6. The fire occurrence probability of two typical points in Inner Mongolia in 2014. (a) and (c) are the daily fire occurrence probability of two typical points in 2014. The grey area is the fire prevention periods, and the grey lines is the value of 0.5; (b) is the kernel density level of historical grassland fire occurrences.**



## 4 Discussion

### 4.1 Temporal and spatial distribution patterns of historical grassland fires

Historical fires were densely distributed in eastern and central Inner Mongolia in space, because typical grassland and meadow grassland were widely distributed here (Fig.2c), which was an area with sufficient fuel. Besides, there were dense traffic routes, and the intensity of human activities is higher than that in western Inner Mongolia. In the northern part of Hulunbuir, the main vegetation type was boreal deciduous coniferous forest (Fig.2c), so grassland fires rarely occurred. In temporal, historical fires was concentrated in spring and autumn (Fig.3), which is closely related to the local climate characteristics. The area was dominated by temperate continental monsoon climate (Xue et al., 2021). In spring, the temperature rose rapidly, the wind speed was high, and the air humidity was low, leading to frequent fires. In autumn and winter, air was dry, increasing the number of fires. However, because of the high latitude, snow covered dead fuel in winter, reducing the probability of fire occurrence (Zhao et al., 2009).

### 4.2 Model comparison in goodness of fitting and predictive ability

Model comparisons have become a hot topic in choosing most suitable prediction models in the last decades (De Vasconcelos et al., 2001; Phelps and Woolford, 2021a; Guo et al., 2016b; Phelps and Woolford, 2021b). In our study, the fitting goodness of GLR model was not competitive with other two models (GWLR and RF) (Table 4). Our results were consistent with previous fire prediction studies (Guo et al., 2016b; Phelps and Woolford, 2021a; Su et al., 2021) which demonstrated that GWLR and RF performs better than GLR in predicting fire occurrences. In addition, the GLR model improperly predicted high probability of fire occurrences during winter at both locations (Fig.6a and c). Temporal distribution of historical fire occurrences showed only fewer fires occurred in winter (Fig.3), suggesting that the prediction ability of GLR model was not ideal in our study area. This may be caused by other crucial environmental variables we did not consider in our study, such as water deficit (Syphard et al., 2018), drought (Rodrigues et al., 2018) etc. needing further to explore in future studies.

Regarding to the GWLR model, it is an improvement of the GLR model by adding a spatial weight matrix to reduce model residuals (Monjarás-Vega et al., 2020), and therefore the goodness of fitting was better than GLR (Table 4). However, the GWLR model also erroneously predicted high probability of fire occurrences during winter and summer at both locations (Fig.6a and c), which was on the contrary to the historical fire statistics (Fig.3). We speculated that GWLR model focused excessively on more significant meteorological variables and ignored the effects of other variables on fire occurrence.

As for the RF model, it had the best goodness of fitting among the three models (Table 4). In addition, the predictions for the two locations generally captured the seasonality of fire probability in our study area, which was higher during spring and autumn, and lower in winter and summer (Shabbir et al., 2020). The predictions were also consistent with the fire prevention periods (Fig.6). Therefore, RF model could be used to predict grassland fire occurrences in our study area.



### 4.3 Factors affecting grassland fire occurrences

Many factors may impact fire occurrences, including vegetation conditions, meteorological/climatic factors, topography and human activities (Muller et al., 2020; Parajuli et al., 2020; Sousa et al., 2021). In our study, meteorological/climatic factors
(daily average specific humidity, daily average wind speed and mean annual temperature) showed higher importance as revealed by the GWLR models. Our results were in line with previous studies (Chang et al., 2013; Guo et al., 2017). Daily average specific humidity had a negative relationship to fire occurrences (Table 2 and Table 3) as with the increase of air humidity, the fuel moisture content also increases, which makes the fuel hard to ignite (Masinda et al., 2021). However, the importance of daily average specific humidity had great spatial variations. More importance was observed in the eastern region
(Fig.4a) since the eastern region was near the sea, the spatial difference of daily average humidity was more obvious.

Generally speaking, the higher the wind speed, the more water vapor produced by plant transpiration will be taken away, and the more sufficient oxygen will be brought to the combustion of fuels (Masinda et al., 2020), thereby increasing the probability of fire occurrences. Therefore, in this study, daily average wind speed was positively related to fire occurrences (Table2).
However, this positive correlation was not spatially uniform with some parts of the central area showed a negative correlation (Fig.4b). We speculated that this may be due to the characteristic of temperate continental monsoon climate. When the wind blows from sea to continental, and brings humid airflow, thereby reducing the probability of fire occurrences.

It is generally believed that temperature positively correlated with fire occurrences (Syphard et al., 2019). On the contrary,
negative relation was showed in our study (Table 2 and Table 3) because the climate characteristics of rain and heat synchronization, and fuel moisture is high and uneasy to be ignite in hot weather. In addition, this negative relation was also spatially non-stationary. In some parts of eastern Inner Mongolia, significant difference was observed (Fig.4c) due to the greater differences of natural geographical features and cultural landscapes in our study area (Rodrigues et al., 2018; Su et al., 2021).


Regarding to anthropogenic factors, the three significant human-related variables were all negatively correlated with fire occurrence as expected (Table 2 and Table 3). Our results were consistent with previous researches (Liu et al., 2012). Historical fires records showed that majority of fires were human-originated. Human activities are more common near roads, settlements and rivers resulting in more fire occurrences (Chang et al., 2013). Therefore, the negative relations of human factors with fire
occurrences were observed. In addition, these relations were also spatially heterogeneous (Fig. 4g, h, i) due to the variations in spatial distribution of roads, rivers and settlements.

As for vegetation conditions, fuel moisture and fuel loads have crucial impacts on fire occurrences (Zhang, 2014; Wu et al., 2020b; Naderpour et al., 2021). We found that the global vegetation moisture index (GVMI) was negatively correlated with





fire occurrence (Table 2 and Table 3), because with the increase of fuel moisture, the fuel is hard to ignite (Su et al., 2021). In addition, significant coefficients of GVMI were located in the eastern region (Fig.4d). This could be explained by the climate constrains hypothesis (Krawchuk and Moritz, 2011) as the fuel in the eastern region is abundant and climate conditions determine fire occurrences. Our results also showed that NDVI was positively correlated with fire occurrences (Table 2 and Table 3) for sufficient fuels is one of the burning conditions (Guo et al., 2017). However, spatially, only a small part of the

western Inner Mongolia had significant NDVI coefficients (Fig.4e). This may support the fuel constrains hypothesis (Krawchuk and Moritz, 2011). Vegetation in the western Inner Mongolia is sparse. Once vegetation coverage increases, it would significantly increase the fire occurrence probability.

Finally, topographic factors also affect fire occurrence patterns (Su et al., 2019; Naderpour et al., 2021). In this study, we found

that the slope and fire occurrence showed a negative correlation in some parts (Table 2 and Table 3). Our results were consistent with the results obtained by Zhang (Zhang, 2014). We believed this was due to the fact that grasslands of Inner Mongolia were mainly distributed on the flat plateau lacking of steep slopes, resulting in a negative correlation between grassland fire occurrences and Slope.

### 4.4 Implications for grassland fire management

From a spatial perspective, our results also showed spatial non-stationarities for various variables in predicting fire occurrences. For examples, the influence of Humi_dy, Wind_dy, GVMI and D_resp on fire occurrences was significant in the eastern Inner Mongolia, and NDVI, D_resp and D_river in the western Inner Mongolia (Fig.4). The spatial non-stationarity of variables may have implications for accurate mapping fire risk zones to fire management based on different environmental factors for various regions, instead of the traditional mapping methods which regard a variable having equal importance across a studied area.

Furthermore, our results showed that RF model performed best in our study area, which should be used to map fire occurrence probability. The spatial pattern of the probability of fire occurrence is of crucial importance in fire management (Sağlam et al., 2008). It could be applied to allocate priority areas for grassland fuel treatments (Wei et al., 2008). From a temporal perspective, fire prevention period in spring and autumn should be adjusted according to local conditions. For example, Point 1 was taken from Hulunbuir region, which has high latitude and insufficient heat. According to RF model prediction, the period of high

fire probability in autumn started and ended earlier, and the period of high fire probability in spring was shorter (Fig. 6a). Therefore, the fire prevention period should be adjusted accordingly.

In our study, we found that majority of grassland fires were human-originated according to the historical fire records, and the peaks of fire occurrences were consistent with the fire prevention periods in Inner Mongolia. Therefore, strict measures should

be taken to prevent fires from occurring, such as forbidding any wildland fire uses during fire prevention periods, building fire monitor systems to timely detect fires so as to extinguish them at the earliest time. In addition, with the increasing intensity of human activities, a large area of grassland was occupied by cultivated land, and the increasing number of agricultural straw



burnings would bring more dangers for grassland. In the circumstance of global warming, the drivers of grassland fire would become more complex. Grassland management strategies must pay more attention to the changes of these drivers and their
relationship with grassland fires to ensure effective fire prevention.

## 5 Conclusions

This study analysed the temporal and spatial characteristics and driving factors of grassland fires in Inner Mongolia Autonomous Region from 2000 to 2018, and compared the prediction accuracy of global logistic regression, geographically weighted logistic regression and random forest models. The spatial distribution of historical fires in Inner Mongolia grassland
was more in the east than in the central than in the west, and the fire high-incidence season was generally same as fire prevention period regulated for Inner Mongolia forest and grassland. The fitting goodness of random forest model was the highest, and geographically weighted logistic regression models also had reference significance to some extent. The influence of 9 drivers selected by geographically weighted logistic regression model on grassland fire was spatially unbalanced. The random forest and geographically weighted logistic regression both showed that meteorological factors were of great
importance to grassland fire. Inner Mongolia had a vast territory, and different areas had different sensitivities to different drivers. Areas with different hydrothermal condition had different fire prevention periods, thus the grassland fire management strategy based on local conditions should be advocated.

*Competing interests.* The contact author has declared that neither they nor their co-authors have any competing interests.

*Acknowledgments.* This research was funded by the National Key Research and Development Program of China Strategic International Cooperation in Science and Technology Innovation Program (2018YFE0207800) and the National Natural Science Foundation of China (grant no. 31971483). We thank the anonymous reviewers for their constructive suggestions to improve the manuscripts.

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
