# Peer review of "Model Comparisons for Predicting Grassland Fire Occurrence Probability in Inner Mongolia Autonomous Region, China"

_Natural Hazards and Earth System Sciences, 2022_

## Referee Comment (RC3)

645

[referee-annotated manuscript omitted]

---

## Author Comment (AC1)

**Editing Certificate**

This document certifies that the manuscript

**Model Comparisons for Predicting Grassland Fire Occurrence Probability in Inner Mongolia Autonomous Region, China**

prepared by the authors

**Chang Chang, Yu Chang, Zaiping Xiong, Xiaoying Ping, Heng Zhang, Meng Guo,Yuanman Hu**

was edited for proper English language, grammar, punctuation, spelling, and overall style by one or more of the highly qualified native English speaking editors at AJE.

This certificate was issued on **January 7, 2022** and may be verified on the AJE website using the verification code **F461-6688-7BA4-09AE-0D20**.

[Figure]

Neither the research content nor the authors' intentions were altered in any way during the editing process. Documents receiving this certification should be English-ready for publication; however, the author has the ability to accept or reject our suggestions and changes. To verify the final AJE edited version, please visit our verification page at aje.com/certificate. If you have any questions or concerns about this edited document, please contact AJE at support@aje.com.

AJE provides a range of editing, translation, and manuscript services for researchers and publishers around the world. For more information about our company, services, and partner discounts, please visit aje.com.

---

## Author Comment (AC3)

Thank you for considering our manuscript. We are thankful to Anonymous Referee #3, for putting plenty of effort into this manuscript and pointing out many instructive, detailed and important modifications needed. We have thoughtfully taken into account these comments. The explanations of what we have changed in response to Referee's concerns are provided point by point. We believe that the comments have been highly constructive and very useful to restructure the manuscript.

1. **L31: This statement is somehow contradicting to the statement in lines 26- 30. Please explain in more detail why and when fires maintain biodiversity and when fires destroy biodiversity. Please consider also the dates of the publication providing this statements.**

   Response:

   We are sorry to make the confusion for you. The proper fire on the grassland will promote nutrient circulation, accelerate ecosystem succession and eliminate inferior tree species, thus promoting the healthy development of the ecosystem and providing a better habitat. However, the excessive and frequent fire in the context of global climate change will destroy the internal structure of grassland ecosystem and damage the habitat of species, which will lead the biodiversity reducing.

2. **L32: Disaster to whom?**

   **How do you define disaster in this context? see e.g. UNU definitions**

   **Disaster to the society using this grasslands? Ecosystems are dynamic, thus they change but this is not disaster.**

   Response:

   Disaster refers to natural variability and extreme events in natural environments that pose a hazard to human life and property

   The frequent and uncontrollable grassland fire will destroy the grassland, leading to the casualties of human and livestock. The disaster here refers to the huge grassland under the fire extreme weather rather than the common fire occurrence due to the ecosystem dynamics change.

3. **L35: But does this support also the ecosystem?**

   Response:

   Although the fire is necessary to ecosystem, currently, in the context of climate change, the frequency of fires has increased, thus the strict fire prevention policy is indispensable to avoid the devastating damage for grassland ecosystem.

4. **L36: How do you define fire risks? What is the difference to threat, hazard or disaster?**

   Response:

   Fire risk refers to the possibility of fire occurrence and the possible consequences in the event of fire.

   Threat means the potential damage.

   Hazard is more formal than risk, it refers to a risk that happens occasionally or cannot be controlled. It usually means that the event is more serious.

   Disaster always refers to the huge devastating events that will bring irreversible consequences, such as earthquake, flood, tsunami, debris flow forest fire and famine, which can be divided into human disaster and nature disaster.

5. **L43: Please provide more details on each of these natural and anthropogenic factors/variables and why these factors/variables are importanten for grassland fire modelling**

   Response:

   Thanks for your suggestion, we will supplement the importance of natural and anthropogenic factors in the Introduction part.

Here is the supplement for manuscript: Climate/weather can influence fire occurrence, with drier environments in some mesic biomes usually displaying more fire activity than wetter ones (Mitchener and Parker, 2005). Climate condition is the key factor for live fuel loading. Topography influences fire occurrence indirectly by contributing to change fuels and their moisture content by changing temperature and water availability (Lafon and Grissino-Mayer, 2007). Higher human population density always increases ignition sources (Syphard et al., 2007) because smoking, outdoor barbecues, vehicle breakdown and other human activities can provide the fire risk for grassland.

6. **L67: Please provide more details on these studies and why they come to their conclusions**
   Response:

   Thanks for your reminder, it's our negligence, and we have replenished the details in the revised manuscript.

   Here is the revised part: For examples, Guo (2016) and Oliveira (2012) both compared the GLR model to RF model in fire predictions, and found that RF model performs better than GLR model. Because RF can generate multiple prediction models at the same time and summarizes the results of the models to improve the accuracy of classification, it is more accurate than GLR (Guo et al., 2016b; Oliveira et al., 2012). Phelps and Woolford employed the logistic regression, bagged classification trees, random forest, neural network to predict the fire occurrence. However, they found all the models were unsuitable for fire occurrence prediction in their study area because all the models overpredicted the number of fire occurrences, which may be due to the bias in their training data samples (Phelps and Woolford, 2021).

7. **L72: Please indicate here clearly the research gaps**
   Response:

   Thanks for your valuable advice.

   Here is the supplement for manuscript: Till now, plentiful scholars always paid more attention to forest fire occurrence and less to grassland fires.

8. **L74: Please try to connect this paragraph with the above one**
   Response:

   Here is the supplement of transition sentence: Grassland fire is extremely active in the temperate grassland of Eurasia (Le Page et al., 2008).

9. **L78: Please explain in more detail how you came to this statement. It is not based on the sentence above or mentioned studies.**
   Response:

   Thanks for your valuable advice.

   We supplement the detail as follows: In order to protect grassland ecosystem, safeguard the life safety of local people and promote the development of husbandry, it is necessary to explore the rules of grassland fire occurrence in Inner Mongolia.

10. **L82: the objectives are clear addressed but they are not well deduced or based on text above. So it is not clear why you see here a knowledged gap and innovative approach in this study**
    Response:

    The gap is that scholars always paid more attention to forest fire occurrence and less to grassland fires.

11. **check this sentence, this is not clear**
    Response:

    This sentence shows the names of terrain units in our study area. We had checked the sentence and there is nothing unclearness. We think it is the cultural difference leading the confusion for you. To avoid the same confusion for more readers, we will delete the sentence in revised manuscript.

12. **considering that you provide here the different types of grassland, it would be an added value if you provide also in the introduction more information on fires and grassland types**

    Response:

    Thanks for the valuable suggestion. However, the study on the fires and grassland types is few, thus lack of references. In addition, the number of fire points on temperate desert grassland in our study is limited, otherwise, we will add the grassland types in our independent variables.

13. **L106: What is the added value of map c)?**

    **Please provide a scale with number such as 100, 250, 500 and 1000 km**

    Response:

    The map(c) is an administrative map, in the result and discussion part, the names of the administrative regions were mentioned, so the map(c) is necessary here.

    Thanks for your advice on the map scale, we have changed the scale of map (b) and map (c) as you described. The revised Figure1 is below:

[Figure]

14. **L109: ... and the source for fire ignition points???**

    Response:

    Thanks for your reminder, we have supplemented the source in the revised manuscript.

15. **L112: (We used the historical grassland fire occurrence records as the dependent variable) Why?**

    Response:

    Because grassland fire occurrence records included the location information of fire points. We want to find the relationship between fire and other environment variables, which is the necessary to enable the fire as a dependent variable.

16. **L117: Why? Please make clear why you chose this method. This aspect was not mentioned before.**

    Response:

    Binomial LR and GWLR models will be adopt in this study. Due to the two binomial models required that the data in a binomial distribution, a certain percentage of random non-fire points should be created to meet the requirements of the binomial LR and GWLR models (Guo et al., 2016a).

We had supplemented this content at the beginning of 2.2.1 in the revised manuscript.

17. **L122: Why did you chose four categories and the mentioned categories? Again the explaination is missing - perhaps you can add the background altready in the introduction - see comments there**

Response:

Thanks for your reminder, we have added the reasons that we chose the 4 categories in the Introduction. Please see the above 5[th] response.

18. **L125: What are the reseon to chose these variables? Please add the missing information to make your study more transpartent**

Response:

We had explained the reasons that why we chose the variables in the following part of the manuscript. Maybe the names of variables here is not suitable, which will bring confusions for readers, and we have deleted the names in the paragraph in the revised.

19. **L127: As you see above on my questions the structure is not clear. You may formulate the first paragraph in such a way that it is clear for the reader that you will explain in more detail.**

Response:

We had adjusted the first paragraph as above comment, the reasons that why we chose the variables had been mentioned in manuscript.

20. **L138: How does this chosen approach influence your results? Did you test this with different random days?**

Response:

We had extracted by random dates for non-fire points for 3 times, and to build models to get the prediction accuracies. Then we find there is a tiny difference on accuracy by the 3 times modeling.

21. **L146:Please check the sentences from line 145 to 150. Did you twice calculated the distance? How did you use the distance as proxies for human activities?**

**Is now the distance to railway, roads and settlment the variable or the population density or both? Why you use both when they are strong correlated? How did you test this?**

Response:

We are very sorry to make you confused. There is something wrong of the statement in the line 145 to 150. We only calculate the Euclidean distance once for each variable. We first calculate the Euclidean distances of railways, roads, rivers and settlements to every raster on the map by ArcGIS. Thus we got 4 Euclidean distances layers of railways, roads, rivers and settlements. Then we overlayed the fire points layers on the 4 layers respectively, and extracted the values according to the location of points as the 4 variables.

As for the variable of Population density, we used the population of every county to generate the raster layer of population density, then overlayed the fire points layers on this layer to extract the population density value as variable. The variable is the population density of the county where the fire point located, rather than the distance to something.

Population density usually has strong correlation with human activity intensity, because the greater the population density in a region, the greater the number of people, which means more human activities, stronger human activity intensity.

In addition, we modified this content in the revised manuscript.

22. **L151: You did not explain why you came up with this factors/variables - Please provide more information**

Response:

We have supplemented the reasons why we chose the 2 variables, below is the supplement in the revised: The fuel is the basic condition of fire burning, and the fuel load will also influence the fire occurrence. Fractional Vegetation Cover (FVC) can be used to indicate the gross of live and dead fuels above the surface. Vegetation moisture content is one of the conditions of vegetation flammability. Global Vegetation Moisture Index (GVMI) can provide direct information on vegetation water content at canopy level.

23. **L151: ???**

Response:

FVC is abbreviation the Fractional Vegetation Cover. We are sorry to miss adding its full name, and will supplement the full name in the revised manuscript.

24. **L152: (Normalized Difference Vegetation Index (NDVI) has the similarity with FVC, so it can be replaced by FVC under some circumstances.) When and why?**

Response:

We are sorry that the statement here is not rigorous, so we changed the expression in the revised: Currently, many methods for calculating FVC by remote sensing have been explored, and one of the more practical methods is to calculate the Normalized Difference Vegetation Index (NDVI) .

25. **L155: How?**

Response:

Here we get NDVI dataset from 2000 to 2018 the Resource and Environment Science and Data Center, then calculate the mean value of the19 layers by Raster Calculator tool in ArcGIS to generate a NDVI layer. Last, we overlayed the NDVI layer and fire point layer to extract the NDVI values as the variables.

26. **L168: I such to formulate this as such : ... is based on Ceccato et al. (2002b)**

Response:

Thanks for your advice, and we modified the sentence according to your advice.

27. **L164: How?**

Response: We use digital elevation model (DEM) as the elevation, downloading the DEM from Geospatial Data Cloud, and processed by spatial correction mosaic, clip, and resample.

28. **L173: How did you consider the different resolutions of your dataset. Did an up- or downscaling? This information is missing**

Response:

All the variable layers had been adjusted to 1000m resolution in data processing.

29. **L182: sentence not clear. This information would be more clear when organised in a diagramm with the variables on the x- and y-axes and the values cells and adding your categories of collinearity with different colours of the cells**

Response:

Thanks for your advice, we have drawn the VIF values of each variable in graph as followed, and added it into revised manuscript. In addition, we did appropriate adjustment on the statement.

[Figure]

Figure 2 The VIF values of every explanatory variables (If VIF>10, there is severe collinearity; 5<VIF<10, there is moderate collinearity; 2<VIF<5, there is mild collinearity (Krebs et al., 2012))

**30. L185: This statement is only clear after learning about the model requirements. Thus, I suggest to move it to the models section.**

Response:

Thanks for your suggestion, it is so reasonable that we have move it to the modeling method section

**31. L190: 20 km?**

Response:

Yes, it's 20km×20km. We modified it in the revised manuscripts.

**32. L190: Please provide more information of the trend analysis in ArcGIS. Which kind of approach was used? There are different possibilities and it is not clear stated in your manuscript**

Response:

Trend analysis is a simple tool in the Geostatistical Analyst in ArcGIS, used to show data in three dimensions. We did not find any approach option needed to be selected. Below are the help document and screenshot of the location of trend analysis tool on the ArcGIS.

[Figure]

[Figure]

You may be interested in mapping a trend, or you might want to remove a trend from the dataset when using kriging. The Trend Analysis tool can help identify trends in the input dataset.

The Trend Analysis tool provides a three-dimensional perspective of the data. The locations of sample points are plotted on the x,y plane. Above each sample point, the value is given by the height of a stick in the z-dimension. A unique feature of the Trend Analysis tool is that the values are then projected onto the x,z plane and the y,z plane as scatterplots. This can be thought of as sideways views through the three-dimensional data. Polynomials are then fit through the scatterplots on the projected planes. An additional feature is that you can rotate the data to isolate directional trends. The tool also includes other features that allow you to rotate and vary the perspective of the whole image, change size and color of points and lines, remove planes and points, and select the order of the polynomial that is to fit the scatterplots. By default, the tool will select second-order polynomials to show trends in the data, but you may want to investigate polynomials of order one and three to assess how well they fit the data.

[Figure]

**33. L192: provide more precise information on your chosen method.**

Response:

We divided a year into 24 parts, every half month as a part. The fire points information includes the months the fires occurred. We reclassified the fire points into the 24 parts according to the months in which they occurred and got the figure.

**34. L156: This sub-chapter would fit better in the introduction than to the method section. Please provide citations for your statements.**

**Only information how you did it in your study should be in the method section. However, here is missing why did you chose a certain software and how you processed the data that you can apply the methods.**

Response:

Thanks for your advice, we have moved some sentences to introduction section, and done some adjustment on this sub-chapter.

In addition, we supplemented the reason that we chose this software and how we processed the data in

the revised manuscript.

35. **L218: which other models did you consider and why you chose this one**

Response:

The GWLR model can be built by the spgwr and GWmodel packages in R, GWR4 software, ArcGIS et al., but GWR4 has a strong operability.

36. **L256: what do you mean with fire preventaion period? Or do you mean fire occurrence period?**

Fire prevention periods is recognized by the authority according to the local climate condition, it's generally considered the high risk period for fire occurrence. In the fire prevention periods, relevant departments will strengthen supervision and implement strict management policies.

37. **L277: check quality of the figure 4**

The figure 4 shows low-resolution in the pdf document by submission system automatic typesetting, and the figure with high-resolution has been uploaded in the submission system.

38. **L297: this should be addressed in the methods section and not in the results**

Response:

Thanks for your advice, we have moved this part about IncNodePurity to methods section in the revised manuscript.

39. **L312: What is the reason that you came up with this conclusion?**

Response:

We are sorry to missing the information about the reason, we have added it in the revised.

Here is the supplement: The RF model has the highest scores of the AUC, $R^2$, and the lowest score of MAE.

40. **L331: check quality of figure 6**

Response:

The problem of figure 6 is same as the figure 4.

In addition, this manuscript had been editing by native English speaking editors, we are very sorry that we can not find any improper expressions in Abstract. Below is the Editing Certificate.

[Figure]

Finally, thank you again for your valuable suggestions and efforts on this manuscript!

**Reference**

Guo, F., Su, Z., Wang, G., Sun, L., Lin, F., and Liu, A.: Wildfire ignition in the forests of southeast China: Identifying drivers and spatial distribution to predict wildfire likelihood, Applied Geography, 66, 12-21, 10.1016/j.apgeog.2015.11.014, 2016a.

Guo, F., Wang, G., Su, Z., Liang, H., Wang, W., Lin, F., and Liu, A.: What drives forest fire in Fujian, China? Evidence from logistic regression and Random Forests, International Journal of Wildland Fire, 25, 10.1071/wf15121, 2016b.

Lafon, C. W. and Grissino-Mayer, H. D.: Spatial patterns of fire occurrence in the central Appalachian mountains and implications for wildland fire management, Physical Geography, 28, 1-20, 10.2747/0272-3646.28.1.1, 2007.

Mitchener, L. J. and Parker, A. J.: Climate, lightning, and wildfire in the national forests of the southeastern United States: 1989-1998, Physical Geography, 26, 147-162, 10.2747/0272-3646.26.2.147, 2005.

Oliveira, S., Oehler, F., San-Miguel-Ayanz, J., Camia, A., and Pereira, J. M. C.: Modeling spatial patterns of fire occurrence in Mediterranean Europe using Multiple Regression and Random Forest, Forest Ecology and Management, 275, 117-129, 10.1016/j.foreco.2012.03.003, 2012.

Phelps, N. and Woolford, D. G.: Guidelines for effective evaluation and comparison of wildland fire occurrence prediction models, International Journal of Wildland Fire, 30, 10.1071/wf20134, 2021.

Syphard, A. D., Radeloff, V. C., Keeley, J. E., Hawbaker, T. J., Clayton, M. K., Stewart, S. I., and Hammer, R. B.: Human influence on California fire regimes, Ecological Applications, 17, 1388-1402, 10.1890/06-1128.1, 2007.